# Blockade of IL-10 Signaling Ensures Mifamurtide Efficacy in Metastatic Osteosarcoma

**DOI:** 10.3390/cancers15194744

**Published:** 2023-09-27

**Authors:** Nicoletta Nastasi, Amada Pasha, Gennaro Bruno, Angela Subbiani, Laura Pietrovito, Angela Leo, Lucia Scala, Lorena de Simone, Gabriella Casazza, Federica Lunardi, Maria Letizia Taddei, Angela Tamburini, Annalisa Tondo, Claudio Favre, Maura Calvani

**Affiliations:** 1Department of Pediatric Hematology–Oncology, Meyer Children’s Hospital IRCCS, 50139 Florence, Italy; nicoletta.nastasi@libero.it (N.N.); amada.pasha@unifi.it (A.P.); gennaro.bruno@unifi.it (G.B.); angela.subbiani@unifi.it (A.S.); angela.tamburini@meyer.it (A.T.); annalisa.tondo@meyer.it (A.T.); claudio.favre@meyer.it (C.F.); 2Department of Experimental and Clinical Biomedical Sciences “Mario Serio”, University of Florence, 50139 Florence, Italy; laura.pietrovito@unifi.it (L.P.); angela.leo@unifi.it (A.L.); marialetizia.taddei@unifi.it (M.L.T.); 3Pharmaceutical Unit, A. Meyer Children’s Hospital, Scientific Institute for Research, Hospitalisation and Health Care, 50139 Florence, Italy; lucia.scala@meyer.it (L.S.); l.desimone@meyer.it (L.d.S.); 4Pediatric Oncology–Hematology Unit, Pisa University Hospital, 56126 Pisa, Italy; g.casazza@ao-pisa.toscana.it (G.C.); f.lunardi@ao-pisa.toscana.it (F.L.)

**Keywords:** osteosarcoma, macrophages, immunotherapy, cytokines, mifamurtide

## Abstract

**Simple Summary:**

Osteosarcoma is a highly aggressive and metastasizing primary bone neoplasm with poor patient survival rates. Mifamurtide is an immunostimulant drug whose clinical efficacy is still debated. Here we identified IL-10 as a new possible target useful to improve mifamurtide effectiveness on metastatic OS. Indeed, we demonstrated that in patients, high levels of IL-10 correlate with mortality. Moreover, the use of anti-IL-10 antibodies causes a significantly increased mortality rate in highest-grade OS cells and lower formation of lung metastases in an in vivo mouse model. These data suggest a possible clinical application of anti-IL-10 antibody and mifamurtide combined treatment as an effective approach for the treatment of metastatic osteosarcomas.

**Abstract:**

Osteosarcoma (OS) is the most common primary malignancy of the bone, highly aggressive and metastasizing, and it mainly affects children and adolescents. The current standard of care for OS is a combination of surgery and chemotherapy. However, these treatment options are not always successful, especially in cases of metastatic or recurrent osteosarcomas. For this reason, research into new therapeutic strategies is currently underway, and immunotherapies have received considerable attention. Mifamurtide stands out among the most studied immunostimulant drugs; nevertheless, there are very conflicting opinions on its therapeutic efficacy. Here, we aimed to investigate mifamurtide efficacy through in vitro and in vivo experiments. Our results led us to identify a new possible target useful to improve mifamurtide effectiveness on metastatic OS: the cytokine interleukin-10 (IL-10). We provide experimental evidence that the synergic use of an anti-IL-10 antibody in combination with mifamurtide causes a significantly increased mortality rate in highest-grade OS cells and lower metastasis in an in vivo model compared with mifamurtide alone. Overall, our data suggest that mifamurtide in combination with an anti-IL-10 antibody could be proposed as a new treatment protocol to be studied to improve the outcomes of OS patients.

## 1. Introduction

OS, also known as “osteogenic sarcoma”, is a highly aggressive and metastasizing primary bone neoplasm derived from malignant mesenchymal cells producing an immature osteoid matrix [1]. It is the most common cancer diagnosed in children and adolescents [2], and it represents 20–40% of all bone tumors [3]. The incidence of OS at diagnosis is 40% higher for males than females [4].

OS differs greatly in its biology and clinical behavior; indeed, patients’ survival rates at 5 and 10 years from diagnosis differ significantly depending on the specific tumor subtype [5]. The incidence of OS is common in the metaphysis of long tubular bones but rare in the spine, pelvis, and sacrum areas [6]. OS has a high tendency to metastasize; the most common sites are the lungs, followed by the bones and, occasionally, the lymph nodes. Some patients develop micrometastases, which cannot be accurately detected by the currently available diagnostic methods. It is well known that the presence of metastases in OS patients is associated with poor prognosis, and it is the primary complexity of tumor therapy [7].

The etiology of OS has not yet been fully clarified; however, some risk factors have been identified and some hypotheses have been formulated on its molecular causes [8,9,10,11,12]. The heterogeneity of OS and the limited understanding of its pathogenesis at present make the diagnosis and therapy really challenging. Currently, the available therapies include intensive multiagent chemotherapy, surgery, and radiation, but often they do not show the expected therapeutic effects, so this points out a real and critical necessity to improve innovative therapeutic approaches, such as immune modulation strategies.

Among immune stimulatory agents, mifamurtide was indicated as one of the most promising for OS treatment. Mifamurtide (muramyl tripeptide ethanolamine (MTP-PE)), a synthetic derivative of muramyl dipeptide (MDP), is the smallest naturally occurring immune stimulatory component of bacterial cell walls [13,14]. Mifamurtide and MDP both stimulate immune responses by binding the nucleotide-binding oligomerization domain-containing protein 2 (NOD2), an intracellular pattern-recognition receptor expressed principally in monocytes, macrophages, and dendritic cells. Mifamurtide activates the nuclear factor (NF)-kB pathway by binding NOD2, which drives an increase in proinflammatory cytokine production (IL-1, IL-6, IL-12, TNF-α) and other serum indicators of immune stimulation (neopterin and C-reactive protein) [15]. Through this pathway, mifamurtide activates macrophages that can target and destroy tumor cells. However, the clinical efficacy of mifamurtide treatment is still debated.

Here, we investigated mifamurtide efficacy in three OS cell lines with increasing features of malignancy through in vitro and in vivo experiments to experimentally test its efficacy and identify new strategies to improve its action.

## 2. Materials and Methods

### 2.1. Study Design

In vivo experiments followed the ARRIVE guidelines and included at least 4 mice per group. All experiments and sample collections were carried out according to the European Union (EU) guidelines for animal care procedures and the Italian legislation (DLgs 26/2014) application of the EU Directive 2010/63/EU (Europe). Experimental animal procedures were approved by the Italian ethical committee of the Animal Welfare Office of the Italian Work Ministry (authorization 796/2021-PR). Mice were housed in a temperature- and humidity-controlled vivarium (12 h dark/light cycle, free access to food and water), and 1 h before the in vivo experiments, mice were acclimatized to the experimental RT.

### 2.2. Cell Culture

MG-63 (CRL-1427), HOS (CRL-1543), and 143B (CRL-8303) human OS cell lines (from (ATCC) were cultured in Eagle’s Minimum Essential Medium (EMEM) supplemented with 10% Fetal Bovine Serum (FBS), 2 mM L-glutamine, 100 U·mL^−1^ penicillin, and 100 μg·mL^−1^ streptomycin. To obtain the 143B complete growth medium, 0.015 mg/mL of 5-bromo-2′-deoxyuridine was added. K7M2 murine OS cells (ATCC, CRL-2836) were cultured in DMEM supplemented with 10% FBS, 2 mM L-glutamine, 100 U·mL^−1^ penicillin, and 100 μg·mL^−1^ streptomycin and were maintained in culture plates at 37 °C in a water-saturated, 5% CO_2_ atmosphere. OS cells were regularly tested for mycoplasma contamination.

### 2.3. Co-Culture and Treatment

OS cells were seeded in multiwell 6 (MG-63 8 × 104/well, HOS, and 143B 5 × 104/well based on their growth rate) and after 24 h were co-cultured with monocytes obtained from buffy coats of healthy volunteers (Collection Center of AOUC Careggi, Florence, Italy) in a seeding ratio of 1:3 (tumor/monocytes) and maintained in a co-culture medium (50% EMEM and 50% RPMI supplemented with 1% heat-inactivated FBS and 1000× macrophage colony-stimulating factor). After 48 h of co-culture, mifamurtide (MEPACT 4 mg Takeda, Takeda Pharmaceuticals, Tokyo, Japan) was added at a concentration of 1 μg/mL after concentration-response experiments performed in MG-63, HOS, 143-B cell lines, and macrophages (Appendix A). After 24 h of treatment, samples were collected for the experiments.

### 2.4. Measurement of Intracellular and Mitochondrial ROS

Intracellular ROS production was measured by DCFH-DA (D6883, Sigma-Aldrich, St. Louis, MO, USA) with a fluorescence microplate reader. OS cells were seeded in 96-well black plates with a density of 5 × 10^4^/well (with 6 parallel wells in each group) and were stained with 10 μM DCFH-DA at 37 °C in the dark for 45 min, then washed with serum-free DMEM (3×). Then ROS levels were determined by an Ensight Multimode Microplate Reader (Perkin Elmer, Waltham, MA, USA) at 488/525 nm.

Mitochondrial ROS activity was measured with the MitoSOX assay (MitoSOX^TM^ Red mitochondrial superoxide indicator, ThermoFisher Scientific Inc., Waltham, MA, USA), a redox-sensitive dye that selectively quantifies mitochondrial ROS. Briefly, OS cells were incubated with 5 μM of MitoSOX reagent working solution for 10 min at 37 °C. Cells were then washed twice with PBS, and fluorescence was determined at 510/580 nm using the MACSQuant Analyzer 10 Flow Cytometer (Miltenyi Biotec^®^, Gladbach, Germany).

### 2.5. Flow Cytometric Analysis

Co-cultured cells and medium were removed, then incubated and stained with different fluorochrome-conjugated antibodies, appropriately diluted, for cytofluorimetric quantification. In particular, the Annexin V-FITC Kit (#130-092-052, Miltenyi Biotec^®^) was used for the identification and enumeration of apoptotic/dead cells. Absolute cell counts were determined by flow cytometry through cell counting beads (CountBrigh absolute counting beads; Invitrogen Corp. Waltham, MA, USA); prior to acquisition, a fixed volume of counting beads cells (10 μL) was added to each sample to obtain the absolute counting cells. Different panels of antibodies were used to discriminate the M1 macrophage population (HLA-DR^+^, CD16^+^, CD80^+^, CD64^+^, CD14^+^, CD86^+^) from the M2 ones (HLA-DR^+^, CD16^+^, CD301^+^, CD163^+^, CD14^+^, CD200^+^, CD206^+^) (Miltenyi Biotec^®^). Stained cells were analyzed using a MACSQuant Analyzer 10 Flow Cytometer (Miltenyi Biotec^®^), and data was processed using Flow Logic 8.7 software (Miltenyi Biotec^®^).

### 2.6. Real-Time PCR

Gene expression analysis was performed using the 2^(−ΔΔCT)^ comparative method of quantification. Briefly, total RNA (1 μg), extracted with the RNeasy Plus Mini Kit (#74134, Qiagen S.p.A., Hilden, Germany), was reverse transcribed using the iScript gDNA Clear cDNA Synthesis Kit (#172-5034, BioRad, Hercules, CA, USA), and samples were previously treated with DNAse according to the manufacturer’s instructions. RNA quantification was performed by NanoDrop 2000/2000c Spectrophotometers (ThermoFisher Scientific). The mRNA level quantification of the target gene was performed in triplicate (GAPDH gene as housekeeping) using TaqMan Universal Master Mix and the automated ABI Prism 7500 Sequence Detector System (ThermoFisher Scientific Inc.). Results were then analyzed by ABI Prism Sequence Detection Systems software, version 1.7 (ThermoFisher Scientific Inc.).

### 2.7. ELISA and AlphaLISA Assay

An ELISA assay was performed to quantify protein expression levels of SOD2 (ab178012, Abcam, Cambridge, UK) and Catalase (ab277396, Abcam) in cell lysates of MG-63, HOS, and 143B cell lines. Both SOD2 and Catalase followed the same manufacturer’s protocol. To each well was added 100 μL of the sample, incubated for 3 h at RT, and then 100 μL of the prepared biotin antibody. The plate was incubated for 1 h at RT, and 100 μL of streptavidin solution was added and incubated for 45 min at RT. Furthermore, add 100 μL of TMB substrate reagent to each well for 30 min at RT. In the end, 50 μL of stop solution was added to each well, and the plate was read immediately at 450 nm. Total and phosphorylated p38 MAPK, STAT3, and NF-κB protein levels were measured using the AlphaLISA technology through the Ensight Multimode Microplate Reader (Perkin Elmer). Briefly, macrophages were collected from co-culture experiments and then lysed with 1X lysis buffer and assayed for phospho and total MAPK (ALSU-TP38-B-HV), STAT3 (MPSU-PTST3-K-HV), and NF-κB (MPSU-PTNFKB-K-HV). 15 μL of acceptor mix were added to 30 μL of cell lysate in a 96-well plate covered with an adhesive film and incubated for 1 h at RT. Furthermore, 15 μL of donor mix were added under subdued light and incubated for 1 h at RT, then the plate was analyzed through a Microplate Reader (Perkin Elmer).

### 2.8. Luminex Cytokines Quantification

IFN-g, VEGF, IL-17A, TGF-b, IL-6, RANTES, IL-10, IL-4, HGF, IL-13, and MCP1 levels (pg/mL) were analyzed by the ProcartaPlex Multiplex Immunoassay (#PPX-10-MXWCWXM, ThermoFisher Scientific), a multi-parameter assay for the quantitative multi-analyte detection of cytokines from cell samples, using Luminex magnetic bead technology. The co-culture medium was collected and processed as the protocol (MAN001694) suggests; 50 µL of magnetic beads were added to each sample, shaken for 30 min at RT, and maintained at 4 °C overnight. The next day, samples were washed (2×), then 25 µL of detection antibody mix (1×) was added and incubated in shackling at RT for 30 min, and to each sample was added 50 µL of streptavidin for 30 min at RT. Samples were washed (2×) and then resuspended in 120 µL of reading buffer before acquisition data with the Luminex^TM^ 100/200 system (ThermoFisher Scientific).

### 2.9. In Vivo Lung Retention Assay

BALB/c mice, 6 weeks old were bought from Envigo (Inotuv, IN, USA). K7M2 cells were stained with a 5 μM CellTracker fluorescent probe (CellTracker green CMFDA #C7025, ThermoFisher Scientific) and intravenously implanted in BALB/c recipient mice by injection of 1×10^6^ cells in 100 μL of PBS in the lateral tail vein. 0.3 μg/mL mifamurtide was administered intraperitoneally (i.p) in 100 μL of PBS; 300 μg of IL-10 neutralization antibody (InVivoMab anti-mouse IL-10 #BE0049, Bio X Cell, NH, USA) were delivered i.p. in 100 μL of InVivoPure pH 7 Dilution Buffer (#IP0070); 300 μg of IgG1 isotype control was delivered i.p. in 100 µL of InVivoPure™ pH 7.0 Dilution Buffer (#IP0070) (vehicle). Treatments were administered 1 h after OS cell injection, and the same protocol was followed for the in vivo treatments with IL-6 (i.p. in 100 μL) and IL-10 (i.p. in 100 μL) alone or in combination with mifamurtide 1 µg/mL. After 24 h, mice were sacrificed, lungs isolated, and dissociated with the Lung Dissociation Kit mouse (#130-095-927, Miltenyi Biotec^®^). In the staining protocol, during IL-6 and IL-10 blockade, Fc blocking reagents were used to avoid false positives and to ensure binding to specific antigens only. The cell suspension was then analyzed through flow cytometric analysis using the MACSQuant Analyzer 10 Flow Cytometer (Miltenyi Biotec^®^) for quantification of K7M2 OS cells.

### 2.10. Statistical Analysis

Data are shown as mean ± SEM. Statistical analysis was carried out by GraphPad Prism 8.0 software (GraphPad, San Diego, CA, USA) by one-way or two-way analysis of variance (ANOVA), followed by the Bonferroni post-hoc test or unpaired student *t*-test; when *p* was <0.05, it was considered significant and described in each figure legend.

## 3. Results

### 3.1. Mifamurtide Is Unable to Exert Its Antitumoral Action in the Most Aggressive Osteosarcoma Cell Lines

The evaluation of mifamurtide efficacy on the induction of tumor cell death on different OS cell lines co-cultured with monocytes showed an induction of apoptosis only in the primary OS cell line, MG-63. Conversely, mifamurtide did not induce apoptosis in more aggressive OS and 143-B cell lines (Figure 1A). Results shown in Figure 1B confirmed a strong reduction of cell viability only in MG-63 treated with mifamurtide compared with metastatic cell lines. MitoSOX staining demonstrated a higher percentage of cells producing ROS and a higher amount of ROS content in the MG-63 cell line than in HOS and 143-B cells at the basal level (Figure 1C,D). Mifamurtide did not change ROS content in MG63 cells; thus, cell death did not depend on ROS production, but it may be relevant in aggressive cells, which showed both ROS decrease and antioxidant activation (Figure 1D,E). Accordingly, SOD-2 and catalase protein expression levels were extremely decreased in MG-63 cell lines after mifamurtide administration, demonstrating an elevated ROS level in these OS cells. Conversely, in HOS and 143B, both SOD-2 and catalase levels were increased after mifamurtide administration, confirming their ability as ROS scavengers (Figure 1E,F). These preliminary results demonstrated that mifamurtide was active only in the MG-63 cell line, while its antitumoral action was ineffective in the more aggressive OS cells.

### 3.2. Mifamurtide Triggers MAPK and STAT3 Signaling Pathways Only in Macrophages Co-cultured with the Less Aggressive MG-63 Cells

It is widely known that mifamurtide, as an immunostimulant drug, acts on macrophage activation and not directly on cancer cells. Indeed, the cytofluorimetric analysis showed that mifamurtide administration increased the M1/M2 ratio in a co-culture assay with all three cell lines, even if the ratio decreased proportionally with cell malignancy, both in treated and untreated conditions (Figure 2A). To verify whether ROS produced by macrophages were involved in the induction of apoptosis of OS cells following mifamurtide treatment, we evaluated ROS levels in macrophages co-cultured with OS cell lines. Results showed that mifamurtide did not increase ROS levels in macrophages co-cultured with tumor cells; inversely, the treatment reduced ROS production in macrophages cultured with all OS cell lines (Figure 2B). It is known that the interaction of mifamurtide with NOD-2 activates cellular pathways such as NF-κB (Nuclear Factor-Kappa B) and MAPK (Mitogen-Activated Protein Kinase) to promote inflammation response and potential tumoricidal effects, leading to an increase in STAT3 phosphorylation, IL-6, and INF-γ production. Data showed that MAPK was truly activated by mifamurtide in macrophages co-cultured with MG-63 and 143-B cell lines, but it caused a significant increase in MAPK phosphorylation only in macrophages co-cultured with MG-63 (Figure 2C). Moreover, mifamurtide induced a significant increase in NF-κB phosphorylation in macrophages co-cultured with MG-63 (Appendix A). Mifamurtide also increases the phosphorylation of STAT3 protein, mainly in macrophages co-cultured with MG-63 cells, but not with statistical relevance (Figure 2D). Finally, mRNA expression of IL-6 and IFN-γ was increased by mifamurtide only in macrophages co-cultured with MG-63, while in the more aggressive HOS and 143B cell lines, the cytokines were decreased following mifamurtide treatment (Figure 2E,F). These data show that, despite the fact that mifamurtide can increase the M1/M2 ratio in co-culture with all the OS tumor cells at different degrees of malignancy, it can activate signaling pathways related to tumoricidal effects only in macrophages co-cultured with the less aggressive MG-63 OS cells.

### 3.3. IL-6 and IL-10 Are Differentially Secreted in Macrophages-Tumor Co-Culture Medium Depending on OS Cell Aggressiveness

The interaction between the immune system and cancer cells is mediated by soluble factors released into the tumor microenvironment. Data revealed that several cytokines, with the ability to promote inflammatory responses against cancer cells, are more expressed by primary tumor cells, and their amount gradually decreases with cell malignancy (Figure 3A). According to literature data, IL-6 and IL-10 cytokines play a major role in the effects brought by macrophage activities in the tumor microenvironment [16]. Our results showed that IL-6 was mainly produced in the culture medium of MG-63 co-cultured with macrophages, and its secretion increased after mifamurtide administration (Figure 3A). The cytokines identified as anti-inflammatory (particularly TGF-β and IL-10) are, instead, higher expressed in cells with more malignant phenotypes HOS and 143-B, proving their contribution in mediating a pro-tumoral effect (Figure 3B). Interestingly, mifamurtide was able to increase the amount of the anti-inflammatory cytokine IL-10 in the coculture medium of the aggressive cell lines HOS and 143-B compared with MG-63, where IL-10 decreased instead after drug administration (Figure 3B). These results suggested that the interaction between macrophages and OS tumor cells is a complex process that is strictly dependent on the biological features of tumor cells and that the tumoricidal effect of mifamurtide is strictly dependent on the cytokine composition of the tumor microenvironment.

### 3.4. Mifamurtide Treatment Causes a Different Trend of IL-10/IL-6 Ratio on OS Cell Lines Depending on the Different Stage of Malignancy

The specific evaluation of the IL-10 and IL-6 expression levels, both in macrophages and in tumor cells, revealed an opposite trend in the IL-10/IL-6 ratio in the co-culture of macrophages with the MG-63 cell line compared with co-culture with HOS and 143B cells following mifamurtide administration. In the macrophage-MG-63 co-culture, the IL-10/IL-6 ratio decreased under the effect of mifamurtide both in macrophages (Figure 4A) and in tumor cells (Figure 4B). Conversely, in macrophages-HOS and macrophages-143B co-cultures, the IL-10/IL-6 ratio increased after mifamurtide administration both in macrophages (Figure 4A) and in tumor cells (Figure 4B). These results agree with the ability of mifamurtide to exert its tumoricidal effect only in the less aggressive MG-63 OS cells, where mifamurtide treatment causes an increase in the anti-tumoral IL-6 at the expense of the pro-tumoral cytokine IL-10.

### 3.5. An Anti-IL-10 Blocking Antibody Triggers the Anti-Tumoral Activity of Mifamurtide on Aggressive OS Cells

Since our results demonstrated that IL-10 and IL-6 cytokines levels are crucial in defining the antitumoral action of mifamurtide, we first assayed cell viability of OS tumor cells treated with mifamurtide in the presence or absence of anti-IL-10 and anti-IL-6 blocking antibodies. Results showed that anti-IL-10 antibody treatment was able to trigger the anti-tumoral effect of mifamurtide in the most aggressive HOS and 143B cell lines, in which mifamurtide treatment alone was not able to affect cell survival. Data on MG-63 cells confirmed, instead, that mifamurtide alone reduced cell survival in less aggressive OS cells (Figure 5A). Conversely, administration of an anti-IL-6 blocking antibody reverted the tumoricidal effect induced by mifamurtide treatment in MG-63 cells, while no change was observed after anti-IL-6 was administered and combined with mifamurtide in the most aggressive HOS and 143B OS cells. These results demonstrated that the administration of an anti-IL-10 antibody was able to trigger the antitumoral effect of mifamurtide in the aggressive OS tumor cells, whereas a single treatment with mifamurtide had no efficacy.

### 3.6. IL-10 Blocking Antibody Converts the Pro-Tumoral IL-10 Cellular Response into a Tumoricidal Action That Synergize with Mifamurtide

IL-10 is associated with the survival, proliferation, and anti-apoptotic activities of various cancers, such as Burkitt lymphoma, non-Hodgkin’s lymphoma, and non-small cell lung cancer [17]. Paradoxically, both inflammatory IL-6 and anti-inflammatory IL-10 signaling depend on the phosphorylation of STAT3, even if they generate nearly opposing cellular responses [18], making it difficult to distinguish the pathways activated by these two cytokines in our experimental setting. Furthermore, literature data showed that IL-10 led to the induction of the transcription factor SOCS3, which leads to the inhibition of signaling triggered by IL-6 [17]. However, it has been demonstrated that shortening activation of STAT3 by IL-10 in the presence of an anti-I-10 antibody changes the IL-10 cellular response into an IL-6-like response [18].

To verify whether the mifamurtide ineffectiveness in aggressive HOS and 143B relied on the blockade of IL-6 signaling caused by a high IL-10 secretion by these tumor cells, we first analyzed the expression of IL-6 in macrophages co-cultured with the three different OS cell lines and treated with an anti-IL-10 blocking antibody alone or in combination with mifamurtide (Figure 5A,B). Results showed that IL-6 levels in macrophages were increased by an anti-IL-10 antibody administration even when co-cultured with the most aggressive HOS and 143B cell lines, and that mifamurtide synergized with the anti-IL-10 antibody in inducing IL-6 expression in macrophages (Figure 5C). To deeper investigate the signaling pathway underlying this effect, cytofluorimetric assays of the transcription factor SOCS3 in macrophages treated with IL-6, IL-10, or their combination with mifamurtide were performed. Results showed that only IL-10 treatment was able to increase SOCS3 expression in macrophages, leading to the neutralization of IL-6 transcription, as shown before. Accordingly, mifamurtide treatment was not able to reverse this effect in the presence of IL-10 (Figure 5D).

Overall, our data confirmed that in OS cells, IL-10 levels are crucial for the efficacy of mifamurtide. Indeed, the IL-10 production by the tumor is able to counteract the tumoricidal effect of the pro-inflammatory cytokine IL-6 secreted by macrophages by inducing, via SOCS3, negative feedback on the IL-6 signaling. This explains the ability of mifamurtide to decrease cell survival only in OS cells where there are low levels of IL-10 (MG-63), while it is ineffective in the most aggressive HOS and 143B cells that have high levels of IL-10. The use of an anti-IL-10 blocking antibody is, in fact, able to trigger the anti-tumoral action of mifamurtide in all OS cells tested.

### 3.7. Administration of Anti-IL-10 Blocking Antibody Synergizes with Mifamurtide in Decreasing Lung Metastasis Dissemination in a Syngeneic OS Model In Vivo

To investigate, in vivo, the efficacy of counteracting OS metastatic dissemination by administration of the anti-IL-10 antibody combined with mifamurtide, we performed a lung retention assay in BALB/c mice injected with the highly metastatic K7M2 osteosarcoma murine cells (Figure 6A). Results showed that anti-IL-10 Ab and mifamurtide alone had no significant effect in reducing lung metastasis dissemination, while the combination of the drugs was able to reduce the number of OS cells disseminated in the lungs of mice (Figure 6B). Moreover, we performed a lung retention assay in BALB/c mice injected with the highly metastatic K7M2 osteosarcoma murine cells after treatment with IL-6 and IL-10, alone or combined with mifamurtide, and we observed a strong reduction of metastases after IL-6 treatment; inversely, IL-10 treatment caused higher production of metastases (Appendix A). These results confirmed the rationale for using an anti-IL10 antibody in combination with mifamurtide to trigger the efficacy of mifamurtide therapy to counteract survival and metastatic dissemination in aggressive OS. Finally, we analyzed the IL-6 and IL-10 levels in the peripheral blood of patients with OS. Results showed lower levels of IL-6 in patients with localized tumors compared withwith metastatic patients, while higher levels of IL-10 were found in metastatic patients (Figure 6B). Furthermore, patients with high levels of IL-10 in PB showed all-cause mortality (Table 1). Altogether, results obtained in the in vivo murine model and in patients with OS confirmed the crucial role played by IL-6 and IL-10 cytokines in defining the anti-tumoral efficacy of mifamurtide treatment, in agreement with data obtained in our in vitro experimental assays.

## 4. Discussion

OS is a type of bone cancer mainly affecting children and young adults that is lethal in over half of cases. The improvement in overall survival remained inactive for over thirty years, especially for recurrent or metastatic cases. Furthermore, due to the genetic complexity of OS, which causes the presence of multiple molecular alterations, targeted therapies have so far found little applicability, but one of the most critical advances in immune-targeted therapy for OS has been mifamurtide therapy. However, within the scientific community, discordant opinions have emerged regarding its efficacy and its possible use in cases of metastatic OS.

Some studies showed a significant increase in the quality of life of OS patients after mifamurtide administration [19]. There was a 25% reduction in the risk of recurrence and a 30% reduction in the risk of death. Moreover, overall survival at 6 years is improved from 70% to 78% with the addition of mifamurtide to standard chemotherapy [20]. On the other hand, some scientists declared the absence of an effective benefit conferred by mifamurtide compared with standard chemotherapy: treatment with mifamurtide can both improve or reduce patient survival and tumor recurrence [21]. In a randomized phase III study in patients affected by OS, the effect of adjuvant chemotherapy alone was compared with chemotherapy and mifamurtide combination, and a negative opinion was expressed on the use of this drug, as no statistically significant differences were observed between the two compared groups. It was observed that the addition of mifamurtide does not truly prolong relapse-free survival. Therefore, the examining commission concluded that the study does not demonstrate the effectiveness of mifamurtide [22]. The clinical evaluation dossier available on mifamurtide reports insignificant data, and several technical commissions have expressed conflicting opinions on the actual clinical relevance of this drug: it has been registered in Europe but was denied approval by the FDA. The ambiguity of these data led us to investigate mifamurtide efficacy on three OS cell lines, characterized by a different degree of malignancy, through in vitro experiments after co-culturing the tumor cells with CD14^+^ cells (monocytes). Mifamurtide exerted its antitumoral effect only in MG-63, while in HOS and 143B, cell viability remained high even after drug administration. Considering these initial results, it was possible to state that there was a variation in drug response depending on the degree of malignancy of the OS cells. HOS and 143B managed to counteract the effect of mifamurtide by finding a way to evade apoptosis. How were cancer cells able to resist the apoptotic process? Especially, could macrophages play an active role in promoting death or survival?

It is well known that in the OS tumor microenvironment, a crucial role is played by both M1 and M2 macrophage populations. Basically, M1 classically activated macrophages have potential tumoricidal functions like phagocytosis, efferocytosis, enhancement of immune response, and direct tumor cell killing effects; instead, M2 alternatively activated macrophages exhibit immunosuppressive features via interaction with various immune effector cells [23,24,25].

Wolf-Dennen et al. observed that metastatic OS cells display a more malignant phenotype via exosomal communication with macrophages. They demonstrated that these metastatic exosomes could induce an M2 phenotype mainly by increasing the expression of transforming growth factor beta 2 (TGFβ2) and IL-10 [26]. Our work reported clear evidence that IL-10 produced by metastatic OS is responsible for the immunosuppressive microenvironment in metastatic cells co-cultured with macrophages. OS would not be the first tumor in which IL-10 has a pathogenetic and metastatic role; in fact, its expression and its ability to shut down the inflammatory response, allowing tumor cells to grow, have been found in a variety of oncological contests, including breast tumors, renal cell carcinomas, non-small cell lung cancers, and melanoma [27,28,29,30,31,32].

The role of this cytokine in inhibiting the inflammatory responses of the host against the tumor and inducing macrophage polarization towards the M2 pro-tumor phenotype has been widely confirmed.

Activated macrophages undergo a strong reprogramming of their cellular metabolism dependent on IL-10. It has been suggested that the anti-inflammatory function of IL-10 acts by controlling essential metabolic pathways, including the signaling of mTOR (mammalian target of Rapamycin), a central regulator of cellular metabolism. IL-10 inhibits the activation of mTORC1 via STAT3 [33].

Furthermore, Ruffell et al. confirmed that IL-10 secreted by M2-type macrophages inhibits the expression of IL-12 by dendritic cells, thereby blocking the response of CD8^+^ T cells [34].

Literature data support the conclusion that IL-10 production within the TME, principally in metastatic tumors, is a common process able to sustain cancer progression and resistance to therapies.

Here, we demonstrated that in metastatic OS cells, mifamurtide treatment alone was ineffective against tumor growth and that the tumor resistance was partly due to IL-10-dependent signaling pathway involvement. The pro-tumor activity of IL-10 relied, at least in part, on the over-expression of SOCS-3 (Suppressor of Cytokine Signaling 3) that in turn inhibited the anti-tumoral IL-6 signaling pathway in macrophages co-cultured with metastatic tumor cells. SOCS-3 is a member of a family of proteins induced by many cytokines (including IL-10), which is generating great interest. SOCS proteins are cytoplasmic proteins inducible either by stimuli that activate the JAK-STAT translation pathway or in a STAT-independent manner and have a broad role in the negative regulation of the response to cytokines and different pro-inflammatory stimuli [35].

Our experimental results showed that administration of an anti-IL-10 antibody triggers the anti-tumoral action of mifamurtide in all OS cells tested, leading to a dramatically decreased survival of tumor cells in macrophage-tumor co-culture in vitro and a drastic reduction of OS cell retention in the lungs of mice in vivo. Moreover, a cohort of patients with metastatic OS showed a high production of IL-10 in circulating monocytes compared withwith patients with a localized OS disease. Inversely, the inflammatory cytokine IL-6 resulted in increased macrophages in patients with localized OS compared with metastatic ones.

The fact that mifamurtide increases IL-10 levels both in macrophages and in aggressive OS tumor cells could be due to the activation of NF-κB as a pro-inflammatory factor, which activates transcription of IL-10 and decreases IL-6 production, leading to an increase in IL-10 levels in metastatic tumor cells. Even if our models cannot precisely mimic the local inflammatory microenvironment of OS, these preclinical results, together with the above-reported clinical data, provide a new understanding of the macrophage role and their inflammatory responses in OS, showing that inhibition of IL-10 signaling pathways could be a useful therapeutic approach that could be used for the development of new cancer immunotherapeutic strategies. Particularly, the combination of IL-10-blocking antibody administration together with other immunostimulatory approaches, such as mifamurtide treatment, could be a promising strategy for aggressive and resistant OS tumor treatment. However, due to the variable biological activity of IL-10 in the tumor microenvironment, it is mandatory to conduct a careful clinical investigation to define the timing and method of anti-IL-10 agent administration. Furthermore, to better understand the IL-10 role in tumorigenesis and the associated immune responses, it is necessary to further study and investigate the mechanisms that regulate IL-10 signaling and expression in tumor sites.

Overall, our results demonstrated that, despite a lack of anti-tumoral efficacy of mifamurtide treatment against aggressive and metastatic OS tumor cells, mifamurtide efficacy can be potentiated by the administration of an IL-10 blocking antibody. One of the mechanisms that we have proposed is an increased production of the anti-inflammatory molecule IL-10 to induce cell death in metastatic cells, and in our consideration, the variability in results between cells is due to the abundance of IL-10 in metastatic cells.

These data pave the way for a novel approach based on the synergic use of mifamurtide together with immunotherapeutic treatments as a novel strategy to delay OS progression, even in its most aggressive forms.

The study requires some further investigation, so future experiments will be highlighted.

## 5. Conclusions

In conclusion, although the use of mifamurtide had a great impact on the treatment of non-metastatic osteosarcoma, much remains to be understood regarding its mechanism of action on more aggressive osteosarcomas. In this work, was highlighted a possible mechanism driven by the action of the anti-IL-10 antibody coupled with mifamurtide which can reverse the resistance of metastatic osteosarcoma to the drug treatment. In fact, the anti-IL-10 antibody with mifamurtide treatment reduced pulmonary metastases in vivo.

## Figures and Tables

**Figure 1 cancers-15-04744-f001:**
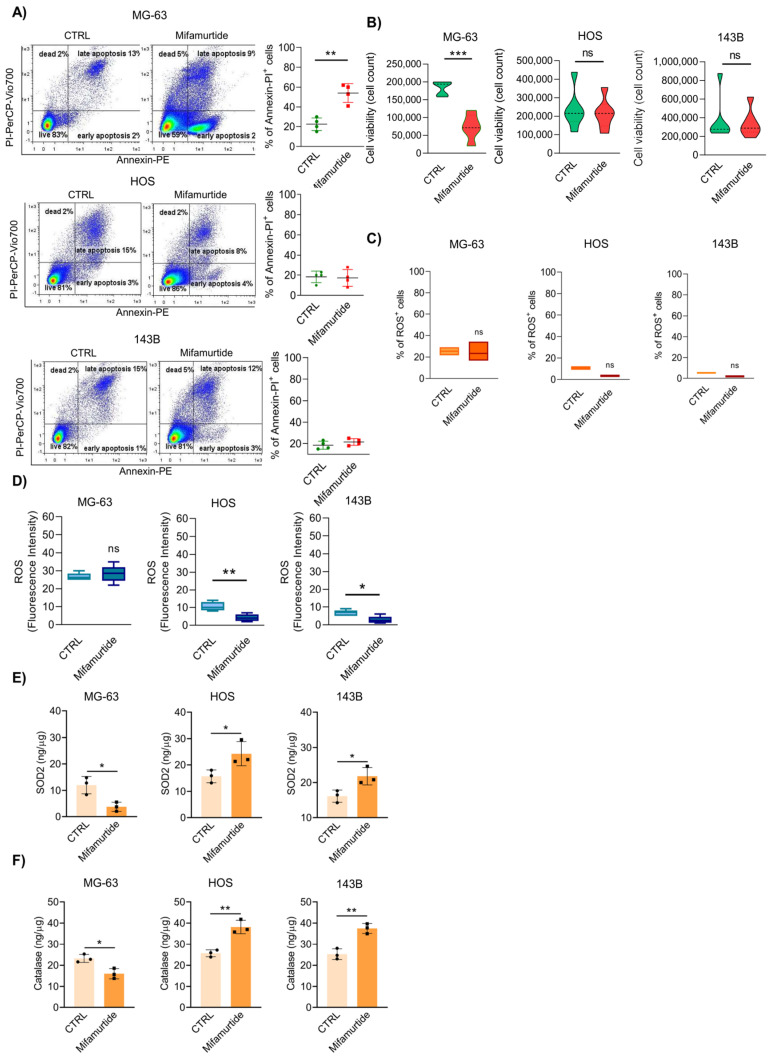
Mifamurtide is unable to exert its antitumoral efficacy in osteosarcoma cell lines. Mifamurtide effect on MG--63, HOS and 143-B OS cells in co-culture with macrophages after 24 h of treatment. (**A**) Representative Annexin V/PI staining plots and relative quantification. (**B**) FACS analysis and relative quantification of live cells count. (**C**) Effect of mifamurtide treatment on oxygen reactive species production (ROS) with MitoSOX^TM^ Red assay. Percentage of ROS^+^ cells are shown. (**D**) Detection of Intracellular ROS generation using the DCFH-DA assay. Fluorescence intensity of ROS production is shown. (**E**) Elisa quantification of SOD-2 and (**F**) Catalase protein expression. Significance was calculated by unpaired Student *t*-test analysis. * *p* < 0.05, ** *p* < 0.01, *** *p* < 0.001 vs. CTRL.

**Figure 2 cancers-15-04744-f002:**
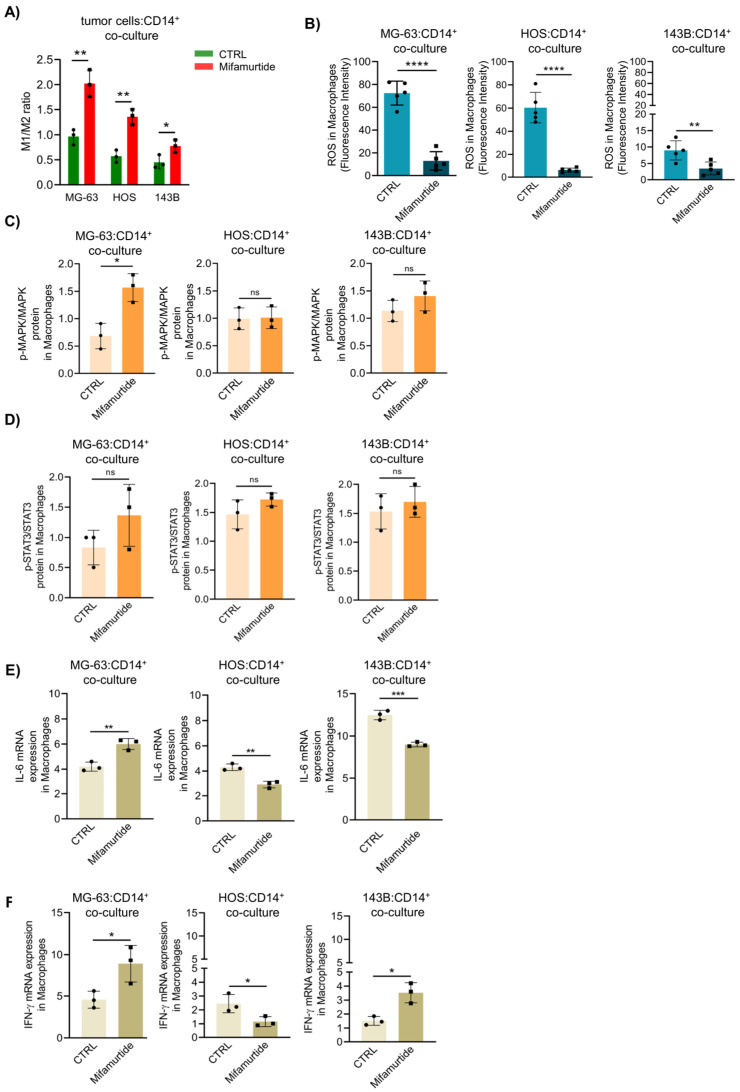
Mifamurtide triggers MAPK and STAT3 signaling pathways only in macrophages co-cultured with MG-63 cells. Mifamurtide signaling on MG-63, HOS and 143-B OS cells in co-culture with macrophages after 24 h of treatment. (**A**) M1/M2 macrophages ratio quantification through FACS analysis. Markers for M1 (HLA-DR^+^, CD16^+^, CD80^+^, CD64^+^, CD14^+^, CD86^+^), and for M2 (HLA-DR^+^, CD16^+^, CD301^+,^ CD163^+^, CD14^+,^ CD200^+^, CD206^+^). For M1 cells were gated on CD45^+^/CD14^+^/NOS2^+^/CD86^+^/CD80^+^. For M2 cells were gated CD45^+^/CD14^+^/CD163^+^/CD206^+^/ARG1^+^. (**B**) FACS analysis and relative quantification of ROS. (**C**) Quantification of p-MAPK/MAPK and (**D**) p-STAT3/STAT3 protein expression levels through AlphaLISA assay in macrophages co-cultured with OS cells. (**E**) IL-6 mRNA and (**F**) IFN-γ mRNA expression levels in macrophages co-cultured with OS cells. Significance was calculated by unpaired Student *t*-test analysis. * *p* < 0.05, ** *p* < 0.01, *** *p* < 0.001, **** *p* < 0.0001 vs. CTRL.

**Figure 3 cancers-15-04744-f003:**
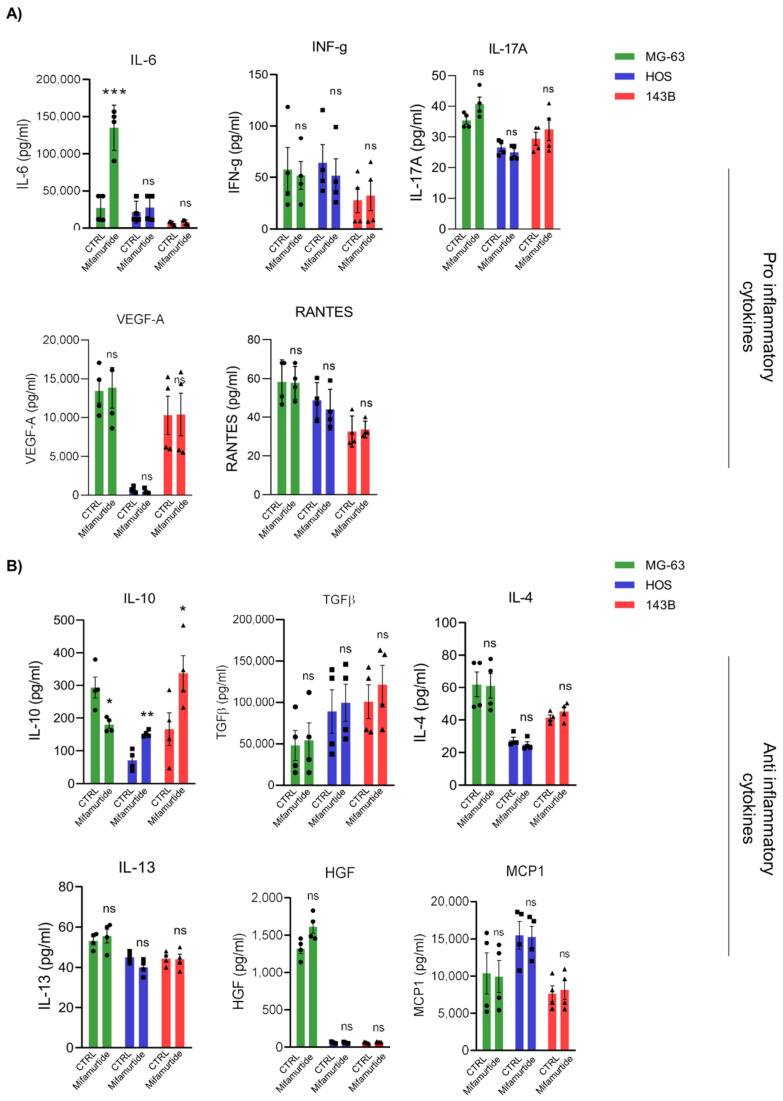
IL-6 and IL-10 are differentially secreted in macrophages-tumor co-cultured medium (**A**,**B**). Quantification of pro-inflammatory (**A**) and anti-inflammatory cytokines (**B**) through Luminex technology on OS cells/monocytes co-culture medium treated and untreated with mifamurtide. Significance was calculated by unpaired Student *t*-test analysis. * *p* < 0.05, ** *p* < 0.01, *** *p* < 0.001 vs. CTRL.

**Figure 4 cancers-15-04744-f004:**
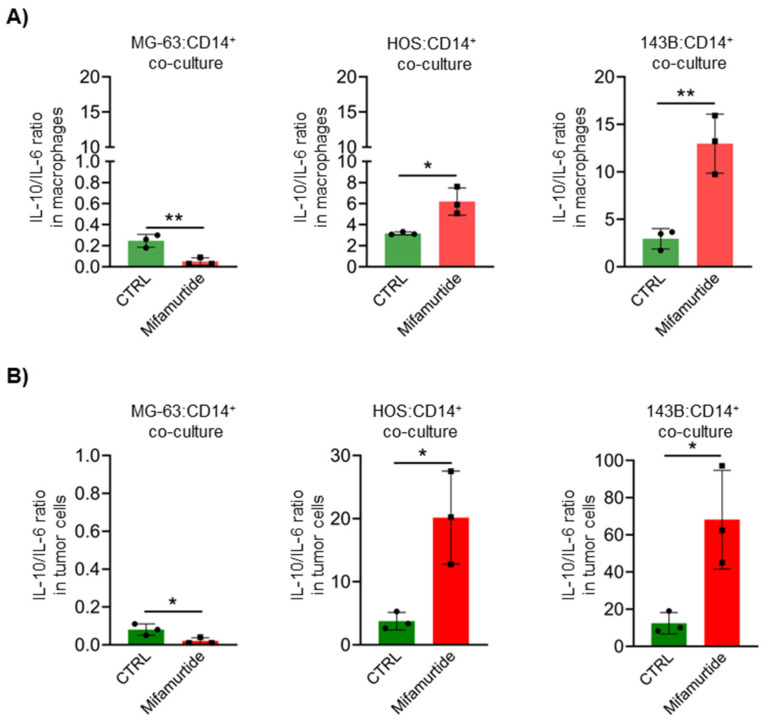
Mifamurtide treatment causes a different trend of IL-10/IL-6 ratio on OS cell lines. IL-10/IL-6 ratio quantification through FACS analysis both in macrophages (**A**) and in tumor cells (**B**) in co-culture conditions treated and untreated with mifamurtide. Significance was calculated by unpaired Student *t*-test analysis. * *p* < 0.05, ** *p* < 0.01 vs. CTRL.

**Figure 5 cancers-15-04744-f005:**
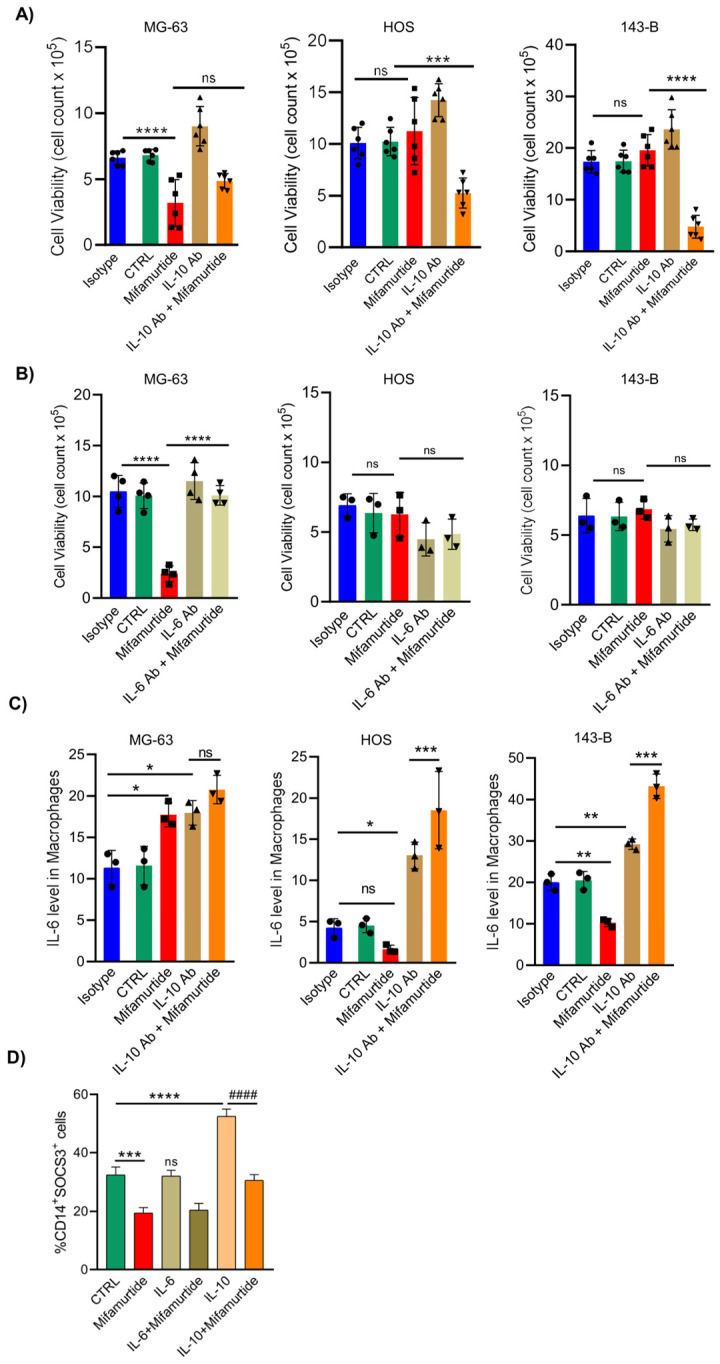
Anti-IL-10 antibody triggers the anti-tumoral activity of mifamurtide. Effect of anti-IL-10 (**A**) and anti-IL-6 antibodies (**B**) treatment alone or in combination with mifamurtide on live cells count measured through FACS analysis where first column corresponds to isotype ctrl. Significance was calculated by unpaired Student *t*-test analysis. *** *p* < 0.001, **** *p* < 0.0001 vs. CTRL. IL-10 blocking antibody converts the pro-tumoral IL-10 cellular response into tumoricidal action. (**C**) Cytofluorimetric analysis of IL-6 expression level in macrophages after IL-10 blocking antibody treatment in co-culture conditions treated and untreated with mifamurtide. (**D**) Cytofluorimetric assay of SOCS3 expression level in macrophages treated with IL-6, IL-10, or their combination with mifamurtide. Significance was calculated by unpaired Student *t*-test analysis. * *p* < 0.05, ** *p* < 0.01, *** *p* < 0.001, **** *p* < 0.0001 vs. CTRL; ^####^
*p* < 0.0001 vs. IL-10.

**Figure 6 cancers-15-04744-f006:**
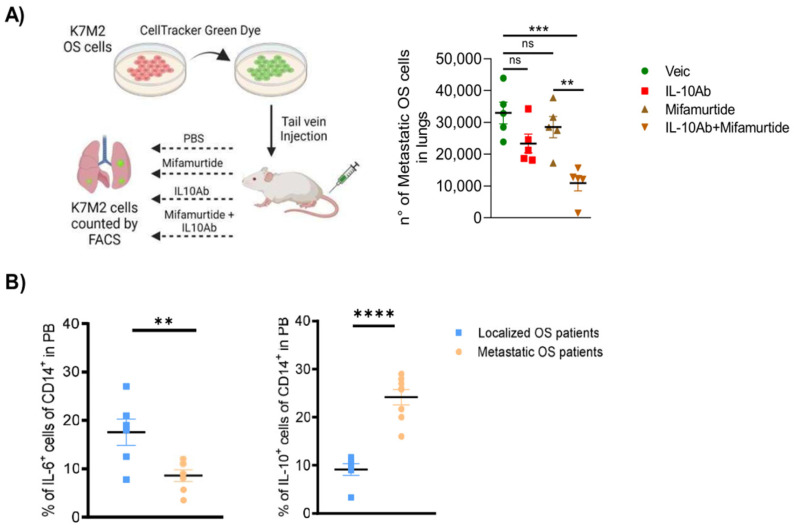
The anti-IL-10 blocking antibody administration synergizes with mifamurtide in decreasing lung metastasis dissemination. (**A**) Experimental scheme of BALB/c mice injected in tail vein with CellTracker green, fluorescent K7M2 OS murine cancer cells and treated with intraperitoneal injections of mifamurtide, an anti-IL10 Ab or their combination. After 24 h, mice were sacrified, lungs isolated and dissociated, and cell suspension was then analyzed through flow cytometric analysis. Number of OS cells were then quantified. (**B**) Quantification of IL-6^+^ and IL-10^+^ cells of CD14^+^ evaluated by FACS analysis on peripheral blood of OS patients with localized or metastatic disease. Significance was calculated by unpaired Student *t*-test analysis. ** *p* < 0.01, *** *p* < 0.001 vs. Vehicle; ** *p* < 0.01, **** *p* < 0.0001 vs. localized OS patients.

**Table 1 cancers-15-04744-t001:** Correlation of IL-6 and IL-10 levels with overall survival in patients. n = number of patients.

Cytokine	ALIVE	DEAD
Low IL-6	n = 12	n = 0
High IL-6	n = 4	n = 8
Low IL-10	n = 12	n = 0
High IL-10	n = 4	n = 8

## Data Availability

The data presented in this study are available in this article.

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
