# Peer review of "Blockade of IL-10 Signaling Ensures Mifamurtide Efficacy in Metastatic Osteosarcoma"

_cancers, 2023, doi:10.3390/cancers15194744_

Round 1

Reviewer 1 Report

In the present study, Nastasi et al. showed that Mifamurtide treatment in combination with IL-10 blockade is promising for the treatment of metastatic osteosarcomas. Below are some concerns.

1.     Material and methods can be improved.

a.     In some experiments, the authors need to count the cell number, but the authors didn’t mention the cell counting method. Did the authors use cell counting beads?

b.     For real-time PCR, what housekeeping gene was used? Did authors treat mRNA samples with DNAase?

c.     Sections 2.7 and 2.8, the authors need to describe the method briefly rather than saying that they were following the protocols.

2.     For the antibody-blocking (IL-10, IL-6 blockade) experiments, samples treated by isotype controls need to be added.

3.     Did authors do Fc receptor blocking when they conduct IL-10 and IL-6 blockade? Because macrophages express Fc receptors, the added antibodies can mediate antibody-dependent cytotoxicity.

4.     It’s better to show the gating strategies of macrophage and tumor cells somewhere in the manuscript.

Reviewer 2 Report

The co-culture experiments lack some control groups, like no-monocytes plus mifamurtide, no-monocytes…… alliteratively the author needs to treat monocytes with mifamurtide first and then co-cultured with tumor cells? What is the concentration of mifamurtide used for in vitro experiments and why? The samples 3-4 is limited. Sample size is too small.

Except ROS, what is the factor in macrophages involved in the apoptosis of OS cells with mifamurtide treatment? IFN-g in macrophages? Need to prove. in another word, what induces OS death when co-cultured with Macrophages? How about the activation of NF-kB signaling with mifamurtide treatment? how did the author detect the protein phosphorylation? What are the markers used for M1 and M2?

What is the difference between MG-63 cell and HOS and 143B cells, because the phenotype is likely to be the opposite. How about the IL-10 levels in Macrophages in the co-culture experiments, better to show the IL-10 alone, not the ratio? Please cite some papers used the ratio IL-10/IL-6. IL-10 can block some pathways like NF-KB signaling, does the IL-10 can affect the inflammation signaling in macrophages?

IL-6 is important, is this cytokine involved in the OS death in the co-culture experiments? Better to make it clear which cytokine works for the system and then test the anti-effect of IL-10. And is IL-6 the key factor for lung metastasis in vivo? What is significance between anti-IL-10 and mifamurtide+anti-IL-10? Could addition with IL-10 cytokine reverse the lung-mets-inhibition of mifamurtide?

the results need improve

Reviewer 3 Report

This research by Nicoletta Nastasi and colleagues, titled “Blockade of IL-10 signaling ensures mifamurtide efficacy in metastatic Osteosarcoma,” highlights role of IL-10 to improve mifamurtide effectiveness on metastatic Osteosarcoma. The study is intriguing, with a well-organized and structured manuscript. The results showcase a commendable comparative analysis between primary and metastatic osteosarcoma cell lines. This research showed lack of anti-tumoral efficacy of mifamurtide treatment against aggressive and metastatic OS tumor cells.
Minor Revision:
1. Certain lines and paragraphs within the manuscript do not lead to any conclusive interpretation. Deciphering the intended message is challenging- Line no. 200-202 and 547-550 (last para of MS).
2. Corrections are required for certain typographical errors, and there is need for improvement in the effective scientific writing.
Major Revision:
1. While numerous parameters were investigated in this manuscript, no mechanistic link was examined to elucidate the reasons for the disparity in study result among the cells. MG-63 (CRL-1427), HOS (CRL-1543) and 143B (CRL-8303) cells (Primary vs Metastatic cells).
2. What is the rationale behind the use of single dose of Mifamurtide.
3. Result 3.1: Showed effect of Mifamurtide on apoptosis, cell viability, ROS. It is suggested that the research team incorporate a minimum of 3 different Mifamurtide dose for the preliminary experiments.
4. Result 3.1 The variation in cell viability arise from the varying sensitivity of different cells to Mifamurtide. Employ diverse treatment dosage to investigate cellular viability.
5. Result 3.1 Result interpretation for the ROS experiment: Does Mifamurtide functions as scavenger of ROS in HOS (CRL-1543) and 143B (CRL-8303) cells and ROS inducer in MG-63 (CRL-1427) cells?
6. Figure 1 is blur; it needs to be improved.

Lack of scientific soundness. 

Round 2

Reviewer 1 Report

NA

Reviewer 2 Report

no more comments